# The effect of end-of-life decision-making tools on patient and family-related outcomes of care among ethnocultural minorities: A systematic review

**Ayah Nayfeh**[1☯]*, **Lesley Gotlib Conn**[1,2☯], **Craig Dale**[2,3☯], **Sarah Kratina**[1], **Brigette Hales**[4], **Tracey Das Gupta**[3,4], **Anita Chakraborty**[4], **Ru Taggar**[4], **Robert Fowler**[1,2,4,5,6☯]

**1** Institute of Health Policy, Management and Evaluation, Dalla Lana School of Public Health, University of Toronto, Toronto, Ontario, Canada, **2** Sunnybrook Research Institute, Toronto, Ontario, Canada, **3** Lawrence S. Bloomberg Faculty of Nursing, University of Toronto, Toronto, ON, Canada, **4** Sunnybrook Health Sciences Centre, Toronto, Ontario, Canada, **5** Interdepartmental Division of Critical Care Medicine, University of Toronto, Toronto, Ontario, Canada, **6** H. Barrie Fairley Professor of Critical Care at the University Health Network, Toronto, Ontario, Canada

☯ These authors contributed equally to this work.
* ayah.nayfeh@mail.utoronto.ca

**Data Availability Statement:** All relevant data are within the manuscript and its Supporting Information files.

## Abstract

### Background

End-of-life decision-making tools are used to establish a shared understanding among patients, families and healthcare providers about medical treatment and goals of care. This systematic review aimed to understand the availability and effect of end-of-life decision-making tools on: (i) goals of care and advance care planning; (ii) patient and/or family satisfaction and well-being; and (iii) healthcare utilization among racial/ethnic, cultural, and religious minorities.

### Methods

A search was conducted in four electronic databases (inception to June 2021). Articles were screened for eligibility using pre-specified criteria. We focused on adult patients (aged ≥18 years) and included primary research articles that used quantitative, qualitative, and mixed-methods designs. Complementary quality assessment tools were used to generate quality scores for individual studies. Extracted data were synthesized by outcome measure for each type of tool, and an overall description of findings showed the range of effects.

### Results

Among 14,316 retrieved articles, 37 articles were eligible. We found that advance care planning programs (eleven studies), healthcare provider-led interventions (four studies), and linguistically-tailored decision aids (three studies) increased the proportion of patients documenting advance care plans. Educational tools (three studies) strongly reduced patient preferences for life-prolonging care. Palliative care consultations (three studies) were

**Funding:** This study was funded by the Sunnybrook AFP Association through the Innovation Fund of the Alternative Funding Plan from the Academic Health Sciences Centres of Ontario; the Department of Critical Care Medicine at Sunnybrook Health Sciences Centre; the Division of Palliative Medicine, Department of Medicine, University of Toronto; the Dalla Lana School of Public Health, University of Toronto; and the Global Institute of Psychosocial, Palliative and End-of-Life Care. Ayah Nayfeh is supported, in part, by funding from the Social Sciences and Humanities Research Council (752-2020-1352). The funders had no role in study design, data collection and analysis, decision to publish, or preparation of the manuscript.

**Competing interests:** The authors have declared that no competing interests exist.

strongly associated with do-not-resuscitate orders. Advance care planning programs (three studies) significantly influenced the quality of patient-clinician communication and health-care provider-led interventions (two studies) significantly influenced perceived patient quality of life.

## Conclusion

This review identified several end-of-life decision-making tools with impact on patient and family-related outcomes of care among ethnocultural minorities. Advance care planning programs, healthcare provider-led interventions and decision aids increased documentation of end-of-life care plans and do-not-resuscitate orders, and educational tools reduced preferences for life-prolonging care. Further research is needed to investigate the effect of tools on healthcare utilization, and with specific patient population subgroups across different illness trajectories and healthcare settings.

## Introduction

End-of-life decision-making tools are used to establish a shared understanding among patients, families, substitute decision-makers, and healthcare providers about medical treatment and goals of care [1]. Several tools have been developed to facilitate making decisions at the end of life and to support facets of the process, such as effective communication and documentation of care plans [2]. When provided with adequate information to make informed decisions for care, patients and/or substitute decision-makers (hereafter referred to as family members) often opt for fewer aggressive treatments–such as cardiopulmonary resuscitation (CPR), hospitalization, and other organ-supporting therapies–when a patient is unlikely to benefit from life-prolonging care [3].

While there is general agreement that the use of end-of-life decision-making tools can assist patients and families to achieve goal-concordant care, the effect of these tools on outcomes of care among ethnocultural minority groups is not well-understood. Prior research has shown a strong association between patient race/ethnicity and increased use of life-prolonging treatments, longer hospital stays, and intensive care units (ICU) as a location of death [4–6]. Racial/ethnic variation in outcomes of care at the end of life may reflect differences in patient preferences based on cultural values, but may also be influenced by challenges related to communication and health literacy, particularly among patients and families with limited English-language proficiency in clinical environments that are English-language dominant [7].

The availability of tools to support end-of-life decision-making with patients and families from ethnocultural minority backgrounds is largely unknown. The effect of such tools on patient and family-related outcomes of care are also poorly understood. To address this gap, we conducted a systematic review to better understand the availability and effect of end-of-life decision-making tools on patient and family-related outcomes of care among ethnocultural minority groups, including: (i) goals of care and advance care planning (ACP), (ii) patient and/or family satisfaction and well-being, and (iii) healthcare utilization.

## Methods

### Search strategy

The population of interest for this review was patients from a racial/ethnic, cultural or religious minority background. We included end-of-life decision-making tools, which we operationally

defined as: acts, interventions, or instruments aimed at supporting patients and/or families in making decisions for end-of-life care, or to assist healthcare providers to better facilitate or engage patients and/or families in the end-of-life decision-making process [2]. This included paper-based or electronic decision aids, ACP programs, communication strategies, healthcare provider-led interventions (e.g., palliative care consults), and educational tools (e.g., videos, booklets). The comparator was usual care/clinical practice, and outcomes of interest focused on goals of care and ACP, patient and/or family satisfaction and well-being, and healthcare utilization.

A search strategy was developed with guidance from a health sciences librarian and conducted in four electronic databases from inception to June 2021: Medline, CINAHL, PsycINFO and Embase. The following keywords and terms were searched independently and in combination: "end of life", "end-of-life care", "decision-making", and "ethnicity/culture". We applied advanced search techniques using different truncations, wild card characters, and "exploding" terms to retrieve related citations. No limitations by language, country of origin, or date were applied to the search. The complete search strategy can be found in S1 Appendix.

The results of multiple searches were merged and reported following the Preferred Reporting Items for Systematic Reviews and Meta-Analysis (PRISMA) guidelines S2 Appendix [8]. RefWorks citation manager was used to remove duplicate items. The reference lists of eligible studies were also manually searched to identify additional articles that were not retrieved in the database search.

## Eligibility criteria

We included primary research articles that employed quantitative (i.e., randomized controlled trials (RCT), observational cohort studies, pre-post studies with or without a control group), qualitative and mixed-methods research designs. To be eligible for inclusion, articles met the following criteria: 1) adult patients (aged ≥18 years); 2) patients from a racial/ethnic, cultural or religious minority background (operationally defined as a minority population within a dominant race/ethnicity, culture or religion); 3) the tool fit our operational definition of end-of-life decision-making: a clinical interaction involving any decision relating to choices for care for a patient with a current or future serious illness (for example, decisions regarding admission to ICU, CPR and interventions to prevent or treat critical illness) [9]; and 4) the study evaluated or tested the effect of a tool.

## Study selection

Two reviewers (AN, SK) independently screened and assessed the eligibility of retrieved articles in Rayyan QCRI [10]. Both reviewers screened the titles and abstracts of the same first 50 publications to calibrate and reconcile any disagreements with the screening process, until consensus was reached. The remaining titles and abstracts were split for screening between the two reviewers. After all titles and abstracts were screened for eligibility, the full text of potentially eligible articles were assessed independently by both reviewers to confirm that inclusion criteria were met. Disagreements about study eligibility were resolved through discussion between the reviewers. Interrater reliability was calculated using Cohen's Kappa statistic for the full text screening phase of potentially eligible articles [11].

## Data collection and charting

A study report form captured key study characteristics such as: article information (year, author, country of origin), research aims, study design, type of tool and characteristics, demographic characteristics, healthcare setting, outcome measures, and findings. The report form

was piloted with a sample of studies to allow for refinement prior to large-scale data extraction. We pre-selected a set of outcome measures based on prior reviews and clinical relevance to the research question [1, 2] to determine the effect of different end-of-life decision-making tools on patient and family-related outcomes of care Table 1. Each outcome measure was used as a label to code and extract data from eligible studies. Extracted data were categorized and synthesized by outcome measure for each type of tool. An overall narrative description of study findings (including point estimates of association and statistical significance) were described to show the range of effects across the different studies. The heterogeneity of eligible studies did not allow for pooling of data to synthesize results. The strength of influence of tools on patient and family-related outcomes of care were determined based on unique outcomes from each individual study.

## Quality assessment

To appraise the quality and risk of bias of individual studies, we used complementary quality assessment tools to generate quality scores for the different methodologies used in eligible studies [12]. The Cochrane Collaboration's tool was used to assess risk of bias of RCT studies (including cluster designs) [13]. The Newcastle-Ottawa Scale was used to assess the quality of observational cohort and case-control studies [14]. The Methodological Index for Non-Randomized Studies (MINORS) tool was used to assess the quality of non-randomized interventional studies [15]. The Agency for Healthcare Research and Quality (AHRQ) tool was used to assess the quality of cross-sectional studies [16]. The 2018 Mixed Methods Appraisal Tool (MMAT) was used to assess the methodological quality of qualitative and mixed-methods research studies, and the 2011 MMAT method was used to generate a numeric score [17].

# Results

## Study selection

A total of 14,316 unique articles were retrieved. The initial screening of titles and abstracts resulted in 77 articles selected for full text review. In total, 37 articles were eligible for inclusion in the review (k = 0.853) (Fig 1).

**Table 1. Patient and family-related outcome measures.**

| Goals of care and advance care planning | Patient and family satisfaction | Healthcare utilization |
|---|---|---|
| • Proportion of patients with documented goals-of-care or ACP discussions, and completion of advance directives.<br>• Proportion of patients with documented do-not-resuscitate (DNR) status.<br>• Proportion of patients with congruence in end-of-life treatment preferences between patients and family members.<br>• Proportion of patients with consistency in end-of-life care between patient wishes and medical orders for life-prolonging treatment.<br>• Proportion of patients with decisions and/or intentions to use life-prolonging treatments at the end of life. | • Patient and family acceptability with tools.<br>• Patient and family satisfaction with tools.<br>• Perceived quality of communication between patients, families and/or healthcare providers.<br>• Perceived pain management and pain severity of patients. | • Proportion of patients receiving life-prolonging treatment at the end of life (i.e., CPR, mechanical ventilation, tube feeding, ICU death, etc.).<br>• Proportion of patients with hospice referrals and hospice care utilization (i.e., length of stay). |

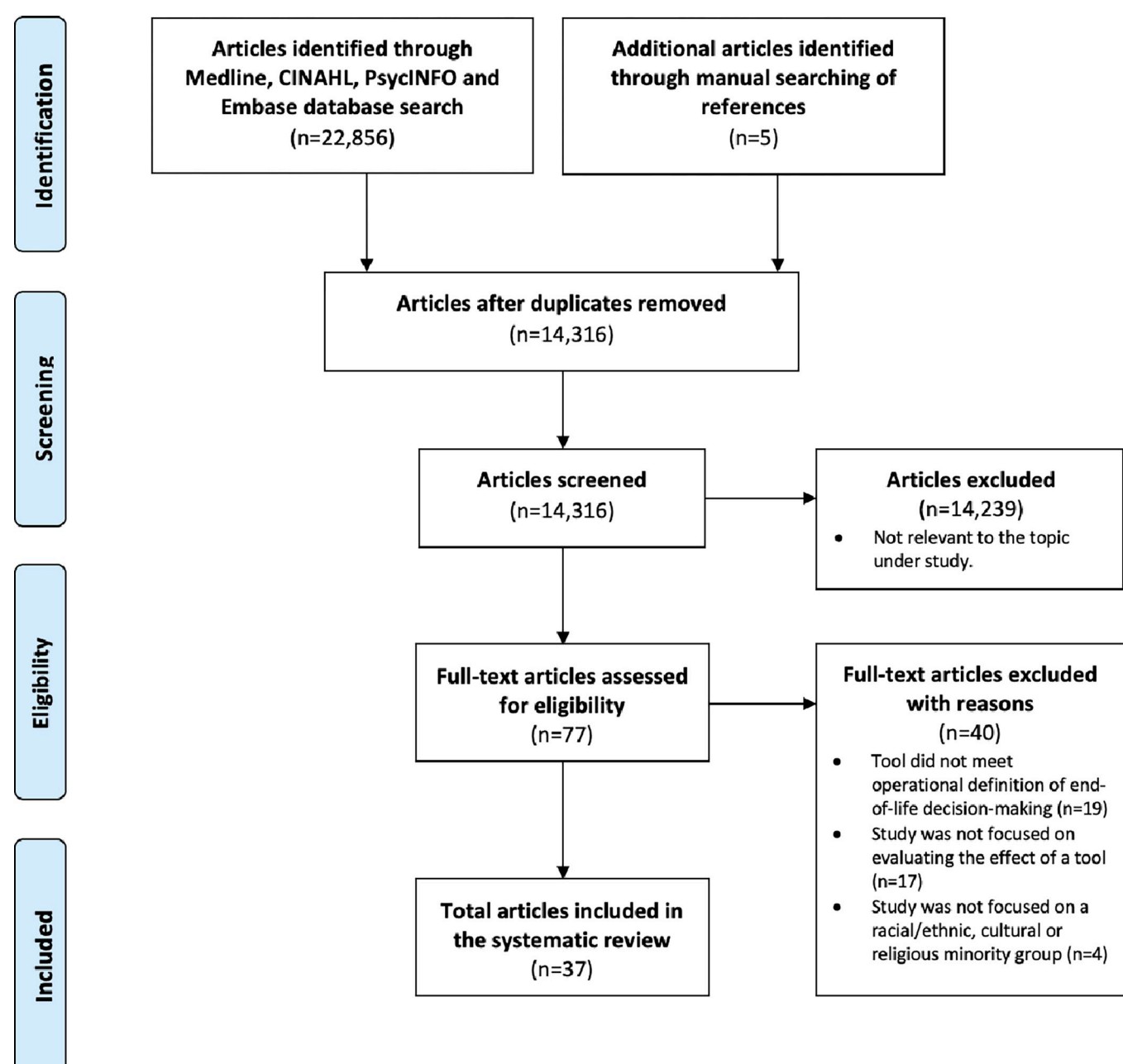

**Fig 1. PRISMA screening flowchart.**

## Characteristics of included studies

**Study setting and population.** Eligible articles were published between 2005 and 2021. Most studies were conducted in the USA (n = 36) and one in Australia (n = 1). Sixteen studies were conducted in community-based settings [18–33], eight in out-patient clinical care units (i.e., dialysis centres, cancer clinics, HIV clinic) [34–41], seven in urban hospitals [42–48] and six in primary care settings [49–54]. Eleven studies focused on a multi-racial/ethnic population [20, 30, 37, 42, 45–49, 51, 53]; 10 on patients of Latino descent [23, 26, 27, 34, 36, 39, 44, 50, 52, 54] and African American descent [18, 19, 21, 24, 32, 35, 38, 40, 41, 43]; two on patients of

Chinese American descent [22, 25]; and one on patients of Chinese Australian [33], Korean American [28], South Asian Indian American [29] and Asian American descent [31]. Study characteristics can be found in S1 Table.

**Types of tools.** Among eligible studies, the types of tools that were identified included ACP programs (i.e., structured ACP programs, mentoring services, toolkits, conversation games, etc.) [21, 22, 24, 26–29, 31, 32, 35, 38, 40, 41, 45, 46]; healthcare provider-led interventions (i.e., palliative care consults, patient navigators, lay health workers, etc.) [34, 37, 42–44, 47–49]; educational tools (i.e., videos, booklets) [18–20, 25, 33, 53, 54]; paper-based and electronic decision aids [30, 50–52]; and communication strategies [23, 36, 39]. Twenty-seven tools were aimed at supporting patients [7, 21, 23–27, 29, 31–34, 36–39, 42–48, 50–53]; five at supporting family members [18–20, 28, 30]; four at supporting both patients and families [35, 40, 41, 49]; and one at supporting healthcare providers [22].

## Quality assessment

Six of the eligible studies were RCT designs, two of which were rated to have an overall low risk of bias [34, 40], three with some concerns [27, 38, 44], and one with a high risk of bias [35]. One study that employed a cluster RCT design was rated to have an overall low risk of bias [18]. Among eight observational cohort studies, four were rated to have a good quality rating [23, 42, 45, 48], three studies had a poor quality rating [43, 46, 47], and one had a fair quality rating [39]. Among 11 studies that employed a non-comparative pre-post study design, six were rated to have a high quality rating (scored 13 to 16 out of a total 16 points) [7, 25, 30, 31, 50, 53] and five had a moderate quality rating (scored 9 to 12 out of a total 16 points) [20, 22, 28, 29, 49]. The five pre-post studies that included a comparator group were rated to have high quality ratings (scored 19 to 24 out of a total 24 points) [19, 37, 41, 51, 52]. Four studies used qualitative (n = 1) and mixed methods (n = 3) research designs. Among these, two studies met 80% of MMAT quality criteria (received 4 out of 5 stars) [21, 32], one study met 60% of MMAT quality criteria (received 3 out of 5 stars) [24], and one study met 40% of MMAT quality criteria (received 2 out of 5 stars) [26]. The two studies that employed cross-sectional research designs were rated to have moderate quality ratings (scored 4 to 7 out of a total 11 points) [33, 36]. The quality assessments for eligible studies can be found in S3 Appendix.

## Outcomes

**ACP programs.** *Proportion of patients with documented goals of care or ACP discussions, advance directives, or advance care plans.* Overall, eleven studies reported an increase in the completion of advance directives among racial/ethnic minority patients after participating in an ACP program (overall moderate quality evidence) [21, 22, 27–29, 31, 32, 35, 38, 45, 46]. Two retrospective chart review studies (rated good [45] and poor quality [46]) and one RCT (with high risk of bias) [35] reported higher completion rates among patients who participated in a structured ACP program (p<0.01). Two RCTs (both had some concerns with risk of bias) also reported higher completion rates among patients who participated in ACP mentoring interventions compared to those who received usual care (odds ratio [OR] = 6.90, 95%CI 1.29–36.66, p<0.05 [27]; OR = 15.9, p<0.01 [38]). Although White patients had higher advance directive completion rates in two studies, a stronger effect was observed among racial/ethnic minority patients post-intervention [45, 38].

Culturally-tailored and faith-based ACP programs appeared to have mild effects on advance directive completion rates among participants in three studies. Two pre-post studies (rated moderate quality) [22, 28] and one qualitative study (which met 80% of quality criteria) [21] reported low overall completion rates (ranging between 3–25%). On the contrary, one pre-

post study (rated high quality) reported a strong influence on advance directive completion rates (71.8% (n = 125/174)) [31]. Mild effects were also reported for an end-of-life conversation game in one pre-post study (rated moderate quality) [32] and one mixed-methods study (which met 80% of quality criteria) (18–41%) [29].

*Proportion of patients with documented DNR status.* Two studies (overall low quality evidence–rated moderate [22] and poor quality [46]) reported a mild increase in DNR orders among racial/ethnic minority patients who used a culturally-relevant toolkit (by 31.8%) [22] and after implementation of a structured ACP program in an urban community hospital (by 3%) [46].

*Proportion of patients with congruence in end-of-life treatment preferences between patients and family members.* Two studies (overall high quality evidence—one RCT had low risk of bias [40] and one pre-post study was rated high quality [41]) reported that structured ACP programs significantly improved congruence in end-of-life treatment preferences between African American patients and family member surrogates at one-week post-intervention ($p<0.05$), but not at three-months post-intervention in the RCT (p = 0.10) [40].

*Proportion of patients with consistency in end-of-life care between patient wishes and medical orders for life-prolonging treatment.* One retrospective chart review (rated good quality) found that a structured ACP program improved consistency between wishes expressed in advance directives and medical orders written to restrict different life-prolonging treatments for racial/ethnic minority patients (ranging between 74 to 96%) [45].

*Proportion of patients with preferences for life-prolonging treatment at the end of life.* One RCT (with low risk of bias) found that preferences for life-prolonging care and CPR were greater among African American patients (n = 10) who participated in a structured ACP program compared to those who received usual care (n = 7) (life-prolonging care: 80% vs. 28.6%; CPR: 90% vs. 57.0%) [41].

*Patient and/or family acceptability with tools.* Two studies (overall moderate quality evidence—one RCT with low risk of bias [40] and one mixed-methods study which met 60% of quality criteria) [24] found that most participants had positive perceptions of structured ACP programs and felt that it helped fill a gap in knowledge around ACP and advance directives. Some patient-surrogate dyads felt that it was difficult to find time to participate in the program and had difficulty facing a patient's incurable illness [40].

*Patient and/or family satisfaction with tools.* Participants in four studies (overall moderate quality evidence—three mixed-methods studies which met 60% [24], 40% [26] and 80% of quality criteria [32] and one pre-post study rated moderate quality [29]) reported being satisfied with structured ACP programs (mean(SD) = 4.81(0.44) on a five-point Likert scale) [24] and an end-of-life conversation game (mean(SD) = 6.21(0.93) [32] and 5.30(1.3) [29] on a seven-point Likert scale).

*Perceived quality-of-communication between patients, families and/or healthcare providers.* Overall, three studies (overall good quality evidence—one RCT with low risk of bias [40] and two pre-post studies rated high [41] and moderate quality [29]) reported that participants were satisfied with the quality of communication after participating in structured ACP programs (intervention: 11.30±1.41 vs. control: 7.52±3.66, $p<0.01$ [40]; intervention: 10.10±2.08 vs. control: 8.14±2.34, $p<0.05$ [41], where 12 = highest quality) and an end-of-life conversation game (mean(SD) = 5.8(1.1) on a seven-point Likert scale) [32].

*Patient and/or family psychological well-being.* Two out of three studies (overall good quality evidence–two RCTs with low risk [40] and some concerns with risk of bias [38]) that evaluated the effect of ACP programs reported an increase in patient subjective well-being ($p<0.05$) [38, 40] and decreased anxiety ($\beta$ = -3.49, p = 0.003) [40] after participating in a structured ACP program and peer-mentoring intervention. On the contrary, one pre-post study (rated high

quality) found no before-after changes in psychospiritual well-being among patients and family member surrogates after participating in a structured ACP program [41].

**Healthcare provider-led interventions.** *Proportion of patients with documented goals of care or ACP discussions, advance directives, or advance care plans.* Overall, five studies reported that a higher proportion of patients completed advance directives after participating in a healthcare provider-led intervention (overall good quality evidence) [34, 37, 44, 48, 49]. One RCT (with low risk of bias) [34] and two pre-post studies (rated good [37] and moderate [49] quality) reported a significant difference in advance directive completion rates among patients who participated in a culturally-tailored patient navigator program (p<0.001) [34] and a lay health worker-led intervention (p<0.001) (compared to those who received usual care) [37], and after participating in a home-based palliative care program (increased from 55% to 90%) [49]. Differences in completion rates between White and African American patients diminished in one pre-post study (rated good quality) after receiving a palliative care consultation [48]. Mild effects were observed in a pilot RCT that had some concerns with risk of bias; however, issues with bias appeared to be rectified in the follow-up RCT [34].

*Proportion of patients with documented DNR status.* Three studies (overall moderate quality evidence—two observational cohort studies rated good [42] and fair [39] quality and one retrospective cohort study rated poor quality [47]) found that palliative and plan-of-care consultations had a strong effect on the proportion and odds of racial/ethnic minority patients with documented DNR orders (adjusted OR = 10.91, p<0.001 [39]; OR = 2.96, 95%CI 2.08–4.22, p<0.0001 [42]; Latino: 70% (n = 625/886) and African American: 65% (n = 724/1113) [47]). However, the odds of documenting a DNR order were significantly lower for racial/ethnic minority patients compared to White patients (adjusted OR = 0.37, 95%CI 0.13–0.99, p = 0.049), and lower for Catholic patients compared to non-Catholic patients (adjusted OR = 0.33, 95%CI 0.13–0.80, p = 0.014) [39].

*Proportion of patients with preferences for life-prolonging treatment at the end of life.* One retrospective cohort study (rated poor quality) found that preferences for life-prolonging care declined substantially among African American patients after receiving a palliative care consultation (from 78.8% to 22.2%, n = 996) [43]. However, African American patients were more likely than White patients to choose aggressive care, both before and after receiving the consultation (relative risk [RR] = 1.17, 95%CI 0.99–1.38 and RR = 1.81, 95%CI 1.70–1.93, respectively).

*Patient and/or family satisfaction with tools.* Caregivers of racial/ethnic minority patients (n = 45) in one pre-post study (rated moderate quality) reported being highly satisfied (mean = 4.8 on a five-point Likert scale) with the overall care provided through a home-based palliative care program [49].

*Perceived patient quality of life.* Two out of three studies (overall good quality evidence—two pre-post studies rated moderate [49] and high quality [37]) reported a significant improvement in patient well-being and quality of life after participating in a home-based palliative care program and a lay health worker-led intervention (p<0.03). On the contrary, one RCT (with low risk of bias) found no statistically significant differences in perceived patient quality-of-life among Latino patients who participated in a culturally-tailored patient navigator program compared to those who received usual care [34].

*Perceived pain management and pain severity of patients.* Two studies that evaluated the effect of healthcare provider-led interventions on pain management and pain severity reported varied findings. One pre-post study (rated moderate quality) found that caregivers of racial/ethnic minority patients were satisfied with relief of patient pain after participating in a home-based palliative care program (mean = 4.2 on a five-point Likert scale) [49]. By contrast, one RCT (with low risk of bias) found no statistically significant differences in pain severity

between patients who participated in a patient navigator program and those who received usual care (p = 0.88) [34].

*Receipt of life-prolonging treatment at the end of life.* A pre-post study (rated moderate quality) found that a home-based palliative care program significantly increased the proportion of patients with no acute care admissions (from 48% to 74%, p = 0.002) [49].

*Proportion of patients utilizing hospice care services.* Three out of five studies (overall good quality evidence—two pre-post studies rated high [37] and moderate quality [49] and one prospective cohort study rated good quality [42]) found that there were greater odds and an increase in the proportion of patients using hospice services after participating in a lay health worker intervention, palliative care consultations, and a home-based palliative care program (adjusted OR = 3.08, 95%CI 2.33–4.07, p<0.0001) [42]. On the contrary, two RCTs (with low risk [34] and some concerns with bias [44]) found that there were no significant changes in hospice use or hospice length of stay among Latino patients with cancer who participated in a patient navigator program compared to those who received usual care.

**Decision aids.** *Proportion of patients with documented goals of care or ACP discussions, advance directives, or advance care plans.* Three pre-post studies [50–52] (all rated high quality) reported that a higher proportion of racial/ethnic minority patients completed an advance directive after using linguistically-tailored decision aids (p<0.03).

*Patient and/or family acceptability with tools.* Two pre-post studies (both rated high quality) found that participants gave high ratings on all acceptability measures after using paper-based ("ease-of-use and understanding": 69.1% vs. 48.7%, p<0.001; personal usefulness in treatment decisions and discussions: 88.6% vs. 75.9%, p<0.001; and general value in care planning: 86.0% vs. 79.0%, p = 0.03) [51] and electronic decision aids (4.2±0.5 for "ease-of-use and understanding" and 4.2±0.8 for "usefulness" on a five-point Likert scale) [30].

*Patient and/or family satisfaction with tools.* One pre-post study (rated high quality) found that racial/ethnic minority older adults (n = 43) were very satisfied with using an ACP website to prepare for ACP and medical decision-making (9±1.9 on a 10-point Likert scale) [30].

*Patient and/or family psychological well-being.* One pre-post study (rated high quality) found that anxiety decreased among diverse older adult participants (n = 10) after using an ACP website (from 23.3% to 16.3% one-week later), although this finding was not statistically significant (p = 0.42) [30].

**Educational tools.** *Proportion of patients with documented goals of care or ACP discussions, advance directives, or advance care plans.* Three studies (overall moderate quality evidence—one cluster RCT with some concerns with risk of bias [18], one pre-post study rated moderate quality [20], and one cross-sectional study rated moderate quality [33]) reported that there were no significant changes in advance directive completion rates among racial/ethnic minority patients who participated in an educational intervention or used educational booklets.

*Proportion of patients with preferences for life-prolonging treatment at the end of life.* Three pre-post studies (all rated high quality) found that educational videos [53, 54] and a group-based educational program [19] strongly reduced the proportion of racial/ethnic minority patients with preferences for life-prolonging care (by 21% to 45%).

*Patient and/or family acceptability with tools.* Two pre-post studies (both rated high quality) found that a high proportion of racial/ethnic minority participants felt comfortable watching educational videos around end-of-life care (between 73% to 93%) [53, 54] and 98.3% felt that the video was "very" or "somewhat" helpful [53]. On the contrary, one cross-sectional study (rated moderate quality) found that a lower proportion of participants (66%, n = 214/325) felt that information received from an ACP education program was useful [33].

**Communication strategies.** *Proportion of patients with documented goals of care or ACP discussions, advance directives, or advance care plans.* One retrospective cohort study (rated good quality) reported that patients who required a language interpreter during an end-of-life discussion were less likely to complete an advance directive compared to those who did not require language interpretation (English-speakers: adjusted OR = 2.6, 95%CI 2.4–2.9; Spanish-speakers: adjusted OR = 1.2, 95%CI 1.1–1.3) [23].

*Proportion of patients with preferences for life-prolonging treatment at the end of life.* One cross-sectional study (rated moderate quality) found that there was strong disagreement with resuscitation among Latino participants when end-of-life discussions were framed with a clause of low probability for survival (adjusted OR = 0.362, 95%CI 0.141–0.925, p<0.05) [36].

## Discussion

This systematic review identified a variety of end-of-life decision-making tools with favourable impacts on patient and family-related outcomes of care. We found that an increased proportion of racial/ethnic minority patients completed advance care plans after participating in ACP programs, healthcare provider-led interventions, and linguistically-tailored decision aids. Educational tools such as videos and group-based education programs reduced racial/ethnic differences and preferences for life-prolonging care (i.e., CPR, mechanical ventilation, tube feeding), and palliative care consultations were strongly associated with DNR orders. Patient and family satisfaction was improved through ACP programs that influenced patient-clinician quality of communication, and healthcare provider-led interventions that influenced perceived patient quality-of-life.

Although prior research has shown that ethnocultural minority patients are less likely to document advance care plans [5, 55, 56], racial/ethnic differences in advance directive completion rates and preferences for life-prolonging care appeared to diminish after using educational tools or participating in palliative care consultations [43, 48, 53]. If preferences at the end of life were based on values specific to certain ethnocultural groups, as some studies suggest [4–6, 55], it is perhaps likely that those decisions would not change after using an end-of-life decision-making tool. The changes observed in this review suggest that disparities in end-of-life decision-making and preferences for care may be influenced by diversity in learning, understanding and communication that can be addressed using different tools.

For example, elderly Latino patients in one pre-post study were more likely to document advance care plans when they were provided with individualized, culturally-competent counselling on advance directives in their preferred language [50]. ACP mentoring interventions–that utilized a more relationship-centered, person-to-person approach to end-of-life decision-making–also effectively increased advance directive completion rates among racial/ethnic minority groups [27, 38]. Acknowledging ethnocultural differences and tailoring approaches to end-of-life decision-making can be a powerful way to enhance trust and participation among specific racialized and minority populations. However, the overall strength and moderate quality of evidence warrants further exploration on the effect of ACP programs to reliably support interpretation of these findings.

Our review found that culturally-tailored [22] and faith-based ACP programs [21, 28] do not have a strong influence on advance directive completion rates among racial/ethnic minorities. Non-tailored end-of-life decision-making tools appear to have a similar effect on outcomes of care among ethnocultural minority groups as compared to other populations. Prior reviews have shown increased documentation of care plans at the end of life after using structured communication tools; reduced patient anxiety after participating in multifaceted ACP programs; and feasibility/acceptability among participants using decision aids [2, 57].

However, we were not able to determine whether the tools identified in our review reduce healthcare utilization and hospitalization for minority patients at a similar rate to White counterparts. The outcomes measured by studies in this review focused largely on goals of care, ACP, and documentation of advance directives. There were few studies that measured outcomes of care further along the end-of-life trajectory, such as healthcare utilization and hospitalization rates, patient/family satisfaction with quality of care, and patient quality-of-life. The overall effect of end-of-life decision-making tools on these outcomes of care remain uncertain. This gap represents an important area for future studies to report through more longitudinal study designs that can assess whether goal-concordant and patient-centered care was indeed achieved.

Although some studies in our review did not find any significant changes in outcomes of care such as DNR status or preferences for life-prolonging treatment, it would be inappropriate to consider these tools as being 'ineffective' based on current Western conceptualizations of quality-of-life and quality-of-care. For instance, our review found that African American patients had greater preferences for life-prolonging care (80% vs. 28.6%) and CPR (90% vs. 57.0%) after participating in a structured ACP program compared to those who received usual care [41]. Although this study involved a small number of participants (intervention n = 10), this finding is consistent with prior research which has found that African American and Latino respondents were more likely to express preferences for intensive care at the end of life and more likely to prefer to die in hospital [58]. This may also explain why some studies in our review reported that White patients had greater advance directive or DNR completion rates and lower preferences for life-prolonging care, both before and after the intervention, compared to racial/ethnic minority counterparts [38, 39, 43, 45]. Follow-up research with an integrated qualitative component could provide broader and/or deeper insight into these findings.

End-of-life decision-making tools should be evaluated in a way that reflects the diversity and complexity of patient wishes at the end of life. Most studies in this review did not distinguish between ethnocultural minority subgroups. For instance, distinctions were not made among Mexicans, Cubans and Puerto Ricans that comprise much of the Latino population in the United States. There were also very few studies in our review that sought complementary qualitative insight or adjusted for other explanatory variables to help explain the outcomes observed. For example, acculturation has been shown to influence attitudes around end-of-life treatment with increased time of residence in the US healthcare context [59]. Religiosity and strong identification with Catholicism–the predominant religion among the Latino American population–has also been shown to be associated with lower ACP engagement and documentation of DNR status [39]. Future studies should seek to investigate and disentangle specific factors that influence the use of end-of-life decision-making tools and decisions for care among specific racial/ethnic, cultural, and religious subgroups. In-depth qualitative studies that use methods such as phenomenology or grounded theory research with specific ethnocultural minority subgroups can also help define outcomes of care to support interpretation of these findings.

We did not identify any studies that evaluated the effect of end-of-life decision-making tools with racial/ethnic minorities in critical care settings. There were only a few studies that focused on urban hospitals and acute care settings, despite nearly 60% of all US and Canadian patients dying in these settings each year [60, 61]. It is unknown whether the end-of-life decision-making tools identified in this review are generalizable across different healthcare contexts. The large majority of included studies focused on patients with chronic diseases or terminal illnesses (such as cancer) that follow a more gradual health decline. It is possible that observed changes in DNR status were more related to the illness trajectory and the amount of time healthcare providers spent with families, rather than specific services or interventions

[43, 47]. Complementary quantitative and qualitative research designs are needed to investigate–in greater depth–the specific components of tools that influence decisions for care among specific patient population subgroups, and how this may differ across different illness trajectories and other healthcare settings.

To our knowledge, this is the first systematic review to examine the availability and effect of end-of-life decision-making tools on patient and families from racial/ethnic, cultural, and religious minority backgrounds. We identified several tools that increased documentation of end-of-life care plans and reduced racial/ethnic differences and preferences for life-prolonging care. Findings from this review represent an important first step towards improving quality-of-care and reducing healthcare disparities among ethnocultural minority populations at the end of life.

## Strengths and limitations

The strength of this systematic review lies in the broad search strategy that was used to identify and describe existing tools to support end-of-life decision-making with patients and families from racial/ethnic, cultural, and religious minority backgrounds. Findings were assessed across a full range of tools, healthcare settings, and patient demographic subgroups. The search was not limited to English language publications and all articles that were retrieved in other languages included English abstracts that helped with the screening process.

A quality assessment was performed for each eligible study and quality scores were presented alongside the summary of findings. More than half of the included studies were rated to have high quality ratings or a low risk of bias. However, there is an important need for high quality studies moving forward around outcomes of care such as quality of life, pain severity, use of hospital services and acceptability of tools. By not limiting our review to one specific type of tool, the heterogeneity of results challenged the analysis and interpretation process. Operational definitions were not always provided in the retrieved studies, and it is possible that our review did not identify potentially relevant articles despite our rigorous search efforts.

We found that there was a paucity of interventions evaluated outside of the US healthcare context and with other racial/ethnic, cultural, or religious minority groups; more than half of eligible studies focused on Latino and African American patients. While it is possible that region-specific terms were missed in our search, which could explain the small number of articles identified outside of the US healthcare context, we attempted to address this limitation by including text word and keyword searches in our strategy to capture studies that describe key concepts around end-of-life decision-making. Therefore, findings from this review may not be fully generalizable to other healthcare settings and contexts outside of the US or with other patient population subgroups. Lastly, we acknowledge our unintended oversight with not registering this review in PROSPERO before the work progressed beyond the point of being able to register it.

## Conclusion

This systematic review identified various end-of-life decision-making tools with impacts on adult patient and family-related outcomes of care among ethnocultural minority populations. We found that ACP programs, healthcare provider-led interventions and educational tools increased documentation of end-of-life care plans and DNR orders and reduced racial/ethnic preferences for life-prolonging care. However, the effect of end-of-life decision-making tools on outcomes of care–such as healthcare utilization and satisfaction with care–remain uncertain. Further research is needed to investigate the effect of tools among specific patient population subgroups across different illness trajectories and healthcare contexts.

## Supporting information

**S1 Appendix. Search strategy for electronic databases.**
(DOCX)

**S2 Appendix. PRISMA checklist.**
(DOCX)

**S3 Appendix. Risk of bias and quality assessments of individual studies.**
(DOCX)

**S1 Table. Characteristics of eligible studies.**
(DOCX)

**S2 Table. Influence of end-of-life decision-making tools on patient and family-related outcomes of care.**
(DOCX)

## Acknowledgments

We thank Erica Lenton (Health Sciences Librarian at the University of Toronto) for her support in developing the search strategy.

## Author Contributions

**Conceptualization:** Ayah Nayfeh, Lesley Gotlib Conn, Craig Dale, Robert Fowler.

**Data curation:** Ayah Nayfeh, Sarah Kratina.

**Formal analysis:** Ayah Nayfeh, Sarah Kratina.

**Funding acquisition:** Ayah Nayfeh, Robert Fowler.

**Methodology:** Ayah Nayfeh, Lesley Gotlib Conn, Craig Dale, Robert Fowler.

**Writing – original draft:** Ayah Nayfeh, Robert Fowler.

**Writing – review & editing:** Ayah Nayfeh, Lesley Gotlib Conn, Craig Dale, Sarah Kratina, Brigette Hales, Tracey Das Gupta, Anita Chakraborty, Ru Taggar, Robert Fowler.

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
