## [Decision Letter · Decision Letter 0]

4 Apr 2022

PONE-D-21-32778The effect of end-of-life decision-making tools on patient and family-related outcomes of care among ethnocultural minorities: a systematic reviewPLOS ONE

Dear Dr. Nayfeh,

Thank you for submitting your manuscript to PLOS ONE. After careful consideration, we feel that it has merit but does not fully meet PLOS ONE’s publication criteria as it currently stands. Therefore, we invite you to submit a revised version of the manuscript that addresses the points raised during the review process.

Please address the reviewers' comments and in particular ensure that the limitations are acknowledged. Please also review the paper for length and reduce the word count by being more concise. 

We look forward to receiving your revised manuscript.

Kind regards,

Lucy Selman, PhD

Academic Editor

PLOS ONE

https://journals.plos.org/plosone/s/file?id=ba62/PLOSOne_formatting_sample_title_authors_affiliations.pdf".

“This study was funded by the Sunnybrook AFP Association through the Innovation Fund of the Alternative Funding Plan from the Academic Health Sciences Centres of Ontario; the Department of Critical Care Medicine at Sunnybrook Health Sciences Centre; the Division of Palliative Medicine, Department of Medicine, University of Toronto; the Dalla Lana School of Public Health, University of Toronto; and the Global Institute of Psychosocial, Palliative and End-of-Life Care. Ayah Nayfeh is supported, in part, by funding from the Social Sciences and Humanities Research Council.”

5. PLOS requires an ORCID iD for the corresponding author in Editorial Manager on papers submitted after December 6th, 2016. Please ensure that you have an ORCID iD and that it is validated in Editorial Manager. To do this, go to ‘Update my Information’ (in the upper left-hand corner of the main menu), and click on the Fetch/Validate link next to the ORCID field. This will take you to the ORCID site and allow you to create a new iD or authenticate a pre-existing iD in Editorial Manager. Please see the following video for instructions on linking an ORCID iD to your Editorial Manager account: https://www.youtube.com/watch?v=_xcclfuvtxQ.

Reviewers' comments:

Reviewer's Responses to Questions

**Comments to the Author**

1. Is the manuscript technically sound, and do the data support the conclusions?

Reviewer #1: Yes

Reviewer #2: Partly

2. Has the statistical analysis been performed appropriately and rigorously? 

Reviewer #1: N/A

Reviewer #2: N/A

3. Have the authors made all data underlying the findings in their manuscript fully available?

Reviewer #1: Yes

Reviewer #2: Yes

4. Is the manuscript presented in an intelligible fashion and written in standard English?

Reviewer #1: Yes

Reviewer #2: Yes

5. Review Comments to the Author

Reviewer #1: Thank you very much for the opportunity to review this well written manuscript on the important topic of the effect of end-of-life decision-making tools on outcomes of care among ethno cultural minorities. The authors have reviewed studies on a variety of end-of-life decision-making tools from a broad perspective: tools for patients, family and healthcare workers, focused on proactive (e.g. ACP) and actual decisions about life prolonging treatments. This broad perspective is relevant, and at the same time results in a diversity of outcomes.

Though the manuscript is interesting, there are some aspects that would benefit from further explanation:

1. Abstract: conclusion could be more informative (see conclusion in the main text)

2. Introduction: the sentence “Prior research has shown a strong association between patient race/ethnicity and use of life-prolonging treatments, hospital stays, and death in intensive care units” needs additional information or rephrasing: more or less use of LPT? Length of hospital stays? As for mortality, do you mean higher mortality or location of dying?

3. Methods, study selection: did both authors screen all titles? Or only the first 50 together and the rest ‘split up’? Does the Kappa statistic reflect the screening of 50 titles or all titles?

4. Methods, data collection: it is not clear to me why data was inductively coded, nor how the coding was done and the codes were subsequently used

5. Methods, quality assessment: the MMAT can be used in reviews that include various designs, both quantitative and qualitative. The authors have chosen to apply MMAT only for qualitative and mixed methods studies, not for the RCTs and other quantitative designs. Furthermore, the quality of studies varied, with 19 studies rated of high quality and 6 of low quality. The authors do not make any distinction in quality in the results. Suggestion to do this, or otherwise discuss the quality in the Discussion section: what is the influence on the interpretation of the results?

6. The results section is rather extensive, due to the summing of outcomes and tools. Please provide less detail, this is already provided in the tables. Focus on the synthesis, whether positive/negative or no effects were found. Now focus often seems on positive results, e.g. on page 10: eleven studies evaluated ACP, of which (only?) 4 had a positive result. From the results section, I have no idea whether these were high quality studies in large groups of patients. You could consider discussing the tools and the effect on outcomes (instead of the other way round), in line with your conclusion.

7. Discussion: can you elaborate on the effectiveness of tools for minorities as compared to other populations (e.g. review Stacey, review Thodé)? And whether tools for minorities seem effective due to adapted communication or a cultural adaptation? Can people with impaired health literacy in general also benefit from the tools found in your review?

8. Discussion: it struck me that all studies were in the USA, and one in Australia. Can you elaborate on that?

Minor comments

a. End of life or end-of-life? Please be consistent

b. Maybe out of scope, but suggestion to acknowledge the health sciences librarian

Reviewer #2: The authors present a systematic review, looking for articles assessing interventions to encourage end of life discussions and decisions. They search from the perspective of wanting to identify studies adapted for patients cultural backgrounds. The methods (apart from the issues raised below) seem sound, and the conduct of the research appears to be rigorous. This review will add to the literature and be of interest to readers from many areas of medicine.

Major issues include:

# The search strategy seems to omit most of the terms associated with treatment limitations. Therefore many studies could have been missed, particularly region specific papers, where a specific limitation phrase is used. Examples of terms related to treatment limitations that could have been included are: Treatment ceiling, POLST, Not for CPR, DNR, goals-of-care etc

# Each theme you have identified should have a statement regarding the strength/reliability of the findings after your synthesis of the data presented the papers. This is in addition to assessing each individual paper for quality/bias etc.

Minor issues include:

# Not registering this study prospectively - eg in PROSPERO

# Your title and abstract should reflect that this study was about adults, not children

# Data are pleural, this should be corrected throughout

6. PLOS authors have the option to publish the peer review history of their article (what does this mean?). If published, this will include your full peer review and any attached files.

Reviewer #1: **Yes: **Irene Jongerden

Reviewer #2: No

---

## [Author Response · Author response to Decision Letter 0]

16 May 2022

Thank you for the opportunity to revise and re-submit our manuscript “The effect of end-of-life decision-making tools on patient and family-related outcomes of care among ethnocultural minorities: a systematic review” (PONE-D-21-32778) for PLOS ONE.

Below we have addressed each of the points provided by the Editor and two reviewers.

Editor’s comments:

Response: Thank you for raising inconsistencies with PLOS ONE’s style requirements. We have revised the title page, main body of the manuscript (i.e., line numbers, heading size, figures/tables) and supporting information to be in accordance with PLOS ONE style guidelines.

Response: Thank you for noting this discrepancy. The funding information section has been updated in the PLOS ONE system. Grant numbers for awards received have been included, where applicable.

Response: We would like to add the following to our financial disclosure statement:

“This study was funded by the Sunnybrook AFP Association through the Innovation

Fund of the Alternative Funding Plan from the Academic Health Sciences Centres of

Ontario; the Department of Critical Care Medicine at Sunnybrook Health Sciences

Centre; the Division of Palliative Medicine, Department of Medicine, University of

Toronto; the Dalla Lana School of Public Health, University of Toronto; and the Global

Institute of Psychosocial, Palliative and End-of-Life Care. Ayah Nayfeh is supported, in

part, by funding from the Social Sciences and Humanities Research Council (752-2020-1352).

Response: Data related to this manuscript is provided as part of the submitted article and supporting information. We have adjusted our Data Availability statement in the PLOS ONE system.

5. PLOS requires an ORCID iD for the corresponding author in Editorial Manager on papers submitted after December 6th, 2016. Please ensure that you have an ORCID iD and that it is validated in Editorial Manager.

Response: The ORCID iD for the corresponding author has been updated in the PLOS ONE system.

Reviewer #1:

1. Abstract: conclusion could be more informative (see conclusion in the main text)

Response: Thank you for this feedback. The following sentence has been added to the conclusion statement in the abstract to provide more information on the findings (page 2, line 49): Advance care planning programs, healthcare provider-led interventions and decision aides increased documentation of end-of-life care plans and do-not-resuscitate orders, and educational tools reduced preferences for life-prolonging care. To stay in line with the 300-word count maximum for the abstract, we have subsequently removed other text from the abstract.

2. Introduction: the sentence “Prior research has shown a strong association between patient race/ethnicity and use of life-prolonging treatments, hospital stays, and death in intensive care units” needs additional information or rephrasing: more or less use of LPT? Length of hospital stays? As for mortality, do you mean higher mortality or location of dying?

Response: Thank you for this comment – we agree that this sentence could use further clarification. We have revised the sentence to read (page 3, line 107): Prior research has shown a strong association between patient race/ethnicity and increased use of life-prolonging treatments, longer hospital stays, and intensive care units (ICU) as a location of death.

3. Methods, study selection: did both authors screen all titles? Or only the first 50 together and the rest ‘split up’? Does the Kappa statistic reflect the screening of 50 titles or all titles?

Response: We have added more information in the manuscript to clarify the process that was followed for study screening/selection (page 6, lines 177). Specifically, the two reviewers independently screened the first 50 articles to calibrate and reconcile any disagreements with the screening process. The remaining titles and abstracts were split for screening for potential eligibility between the two reviewers. After all titles and abstracts were screened for potential eligibility, the full text of potentially eligible articles were assessed independently by both reviewers to confirm that inclusion criteria were met. The Cohen’s Kappa statistic represents interrater reliability for the full-text screening phase of potentially eligible articles. 

4. Methods, data collection: it is not clear to me why data was inductively coded, nor how the coding was done and the codes were subsequently used.

Response: We have adjusted the language to better describe the process that was followed for data analysis and how codes were used to analyze the data (page 6, line 190): 

Each outcome measure was used as a label to code and extract data from eligible studies. Extracted data were categorized and synthesized by outcome measure for each type of tool. An overall narrative description of study findings (including point estimates of association and statistical significance) were described to show the range of effects across the different studies. The heterogeneity of eligible studies did not allow for pooling of data to synthesize results.

5. Methods, quality assessment: the MMAT can be used in reviews that include various designs, both quantitative and qualitative. The authors have chosen to apply MMAT only for qualitative and mixed methods studies, not for the RCTs and other quantitative designs. Furthermore, the quality of studies varied, with 19 studies rated of high quality and 6 of low quality. The authors do not make any distinction in quality in the results.

Suggestion to do this, or otherwise discuss the quality in the Discussion section: what is the influence on the interpretation of the results?

Response: Thank you for this comment. We selected quality assessment tools based on the recommendations of Zeng et al. (2015) for different research methodologies. Based on their assessment, the authors recommended using MMAT for qualitative and mixed method research designs; quantitative research designs (such as case-control, observational cohort, and non-randomized intervention studies) were given other recommendations. We highlighted the recommended quality assessments tools on page 7, line 212 of the manuscript.

Thank you also for the suggestion to reflect on the quality of study findings in the discussion section. We have incorporated statements around the overall quality of evidence to support interpretation of results in the discussion section (page 18, line 483; page 19, line 515).

Reference: Zeng X, Zhang Y, Kwong JSW, Zhang C, Li S, Sun F, et al. The methodological quality assessment tools for preclinical and clinical studies, systematic review and meta-analysis, and clinical practice guideline: a systematic review. J Evid Based Med. 2015;8(1):2–10.

6. The results section is rather extensive, due to the summing of outcomes and tools. Please provide less detail, this is already provided in the tables. Focus on the synthesis, whether positive/negative or no effects were found. Now focus often seems on positive results, e.g. on page 10: eleven studies evaluated ACP, of which (only?) 4 had a positive result. From the results section, I have no idea whether these were high quality studies in large groups of patients. You could consider discussing the tools and the effect on outcomes (instead of the other way round), in line with your conclusion.

Response: Thank you for this important comment. We agree that the results section could be presented in a more concise manner. As suggested, we re-organized the results section such that the findings are organized by tools rather than by outcomes. This is more in line with our conclusion statement. This approach also allows readers to more easily comprehend the influence of different tools on patient and family-related outcomes of care.

We have also revised the results section to be more concise by focusing primarily on the synthesis of findings and overall strength/reliability of evidence. In doing so, the results section was cut down by almost four pages. Additional details that support interpretation of the findings can be found in the supporting tables.

7. Discussion: can you elaborate on the effectiveness of tools for minorities as compared to other populations (e.g. review Stacey, review Thodé)? And whether tools for minorities seem effective due to adapted communication or a cultural adaptation? Can people with impaired health literacy in general also benefit from the tools found in your review?

Response: Thank you for this suggestion to enhance the discussion section of this manuscript. We have reflected on our findings in the context of prior reviews and discussed potential application and generalizability to other populations as well. Specifically, we have included the following paragraph on page 18, line 486:

Our review found that culturally-tailored [23] and faith-based ACP programs [22,29] do not have a strong influence on advance directive completion rates among racial/ethnic minorities. Non-tailored end-of-life decision-making tools appear to have a similar effect on outcomes of care among ethnocultural minority groups as compared to other populations. Prior reviews have shown increased documentation of care plans at the end of life after using structured communication tools; reduced patient anxiety after participating in multifaceted ACP programs; and feasibility/acceptability among participants using decision aides [2,58]. However, we were not able to determine whether the tools identified in our review reduce healthcare utilization and hospitalization for minority patients at a similar rate to Caucasian counterparts.

8. Discussion: it struck me that all studies were in the USA, and one in Australia. Can you elaborate on that?

Response: Although our search strategy retrieved articles around end-of-life decision-making beyond the USA (including from the UK, Canada, France, Australia, Saudi Arabia, Iran, China), the majority of these studies did not meet our criteria for inclusion in the review. The main reason for exclusion was that the study did not evaluate or test the effect of the tool or intervention, or the study did not focus on a racial/ethnic, cultural or religious minority group. We anticipated that most studies would be based in the US healthcare system considering growing recognition and conversations around racial/ethnic disparities in health.

It is possible that there were region-specific terms that were missed in our search. We attempted to address this limitation by including text word and keyword searchers in our strategy to capture studies that describe key concepts around end-of-life decision-making. For example, we searched for articles that included the following terms within 3 or 4 words of each other:

(life adj3 (prolong* or sustain* or support*)).tw,kf. 

(care adj3 (terminal or palliative or intensive or critical or hospice)).tw,kf. 

((ill or illness*) adj3 (terminal* or serious* or critical*)).tw,kf. 

((withdraw* or withhold* or refus* or discontinu* or remov* or terminat*) adj3 (treatment* or care or therap* or intervention* or medical*)).tw,kf. 

((plan* or decision*) adj4 (care or medical or treatment)).tw,kf. 

Despite casting a wide net in our search, however, there may have been region-specific terms that were missed in our search. This is an important limitation that we have addressed in the manuscript (page 22, line 598).

9. End of life or end-of-life? Please be consistent

Response: Thank you for raising this comment. We are aware that some journals prefer to use the terms “end of life” and “end-of-life care” differently. We have attempted to adhere to the language rule that specifies hyphens when the term is being used as a descriptor for a process, event, etc. and no hyphens when it is used by its own. For example(s): We wish to improve end-of-life care. We wish to improve care at the end of life.

10. Maybe out of scope, but suggestion to acknowledge the health sciences librarian

Response: This is a wonderful suggestion – we have included an acknowledgement of the health sciences librarian in the manuscript.

Reviewer #2:

10. The search strategy seems to omit most of the terms associated with treatment limitations. Therefore many studies could have been missed, particularly region specific papers, where a specific limitation phrase is used. Examples of terms related to treatment limitations that could have been included are: Treatment ceiling, POLST, Not for CPR, DNR, goals-of-care etc.

Response: Thank you for this important comment. We are fairly confident that the text word and keyword function in our search strategy captured studies related to goals of care, POLST, CPR and DNR. For example, our search included:

(goal* adj3 care).tw,kf [this would have captured any article where the words ‘goal’ and ‘care’ were within 3 words of each other].

resuscitation/ or cardiopulmonary resuscitation/ or respiration, artificial/ or resuscitation orders/ [this would have captured any article with terms related to do-not-resuscitate (DNR) and CPR]

(life adj3 (prolong* or sustain* or support*)).tw,kf [this would have captured any article related to Physician Orders for Life Sustaining Treatment (POLST)]

Certainly, there may be other region-specific terms that were missed in our search. This is an important limitation that we have addressed in the manuscript (page 22, line 598).

11. Each theme you have identified should have a statement regarding the strength/reliability of the findings after your synthesis of the data presented the papers. This is in addition to assessing each individual paper for quality/bias etc.

Response: Thank you for this comment – we have adjusted the results section such that the focus is on the synthesis of findings and overall quality of evidence. A statement around the overall strength and reliability of findings has been included for every outcome measure under each tool (in addition to each individual quality/risk of bias assessment).

12. Not registering this study prospectively - eg in PROSPERO

Response: We acknowledge our unintended oversight with not registering this review in PROSPERO. As this systematic review is complete, we were not able to register the review retrospectively. We acknowledge this oversight in the manuscript that the work progressed beyond the point of being able to register it by the time we attempted to register. This has been addressed in the limitations section of the manuscript (page 22, line 604).

13. Your title and abstract should reflect that this study was about adults, not children

Data are pleural, this should be corrected throughout.

Response: Thank you for this suggestion. We have added information in the abstract to indicate to readers that this review focuses on the adult patient population. Grammar corrections have also been made throughout the manuscript.

---

## [Decision Letter · Decision Letter 1]

20 Jul 2022

The effect of end-of-life decision-making tools on patient and family-related outcomes of care among ethnocultural minorities: a systematic review

PONE-D-21-32778R1

Dear Dr. Nayfeh,

We’re pleased to inform you that your manuscript has been judged scientifically suitable for publication and will be formally accepted for publication once it meets all outstanding technical requirements.

Kind regards,

Lucy E. Selman, PhD

Academic Editor

PLOS ONE

Additional Editor Comments (optional):

Thank you for your revision, which we are pleased to accept. The reviewer highlighted the spelling of 'aid' - please amend to 'aid' throughout.

Reviewers' comments:

Reviewer's Responses to Questions

**Comments to the Author**

1. If the authors have adequately addressed your comments raised in a previous round of review and you feel that this manuscript is now acceptable for publication, you may indicate that here to bypass the “Comments to the Author” section, enter your conflict of interest statement in the “Confidential to Editor” section, and submit your "Accept" recommendation.

Reviewer #1: All comments have been addressed

2. Is the manuscript technically sound, and do the data support the conclusions?

Reviewer #1: Yes

3. Has the statistical analysis been performed appropriately and rigorously? 

Reviewer #1: Yes

4. Have the authors made all data underlying the findings in their manuscript fully available?

Reviewer #1: Yes

5. Is the manuscript presented in an intelligible fashion and written in standard English?

Reviewer #1: Yes

6. Review Comments to the Author

Reviewer #1: Compliments on the revised version! It has improved significantly. Just one minor comment: as far as I know, it is 'decision aids' rather than 'decision aides'.

7. PLOS authors have the option to publish the peer review history of their article (what does this mean?). If published, this will include your full peer review and any attached files.

Reviewer #1: **Yes: **Irene Jongerden

---

## [Editor Report · Acceptance letter]

27 Jul 2022

PONE-D-21-32778R1 

The effect of end-of-life decision-making tools on patient and family-related outcomes of care among ethnocultural minorities: a systematic review 

Dear Dr. Nayfeh:

I'm pleased to inform you that your manuscript has been deemed suitable for publication in PLOS ONE. Congratulations! Your manuscript is now with our production department. 

Kind regards, 

on behalf of

Dr Lucy E. Selman 

Academic Editor

PLOS ONE